# Properties of Selected Alternative Petroleum Fractions and Sustainable Aviation Fuels

**Hugo Kittel \*** , **Jiří Horský and Pavel Šimáček**

Department of Petroleum Technology and Alternative Fuels, Faculty of Environmental Technology, University of Chemistry and Technology, Technická 1905, 166 28 Prague, Czech Republic
\* Correspondence: hugo.kittel@vscht.cz; Tel.: +420-220444236

**Abstract:** With regard to speed, comfort, and a dense network of destinations, the popularity of air transport is on the rise. For this reason, jet fuel is a commodity with rapidly growing consumption and interesting refinery margins. At the same time, however, it is becoming a focus of attention in terms of reducing negative environmental impacts. As a response to these trends, it will be necessary to coprocess alternative petroleum fractions with sustainable aviation components in oil refineries. Six alternative jet fuel samples of different origin were used to investigate their jet fuel-specific properties, that is, aromatics (from 0 to 59.7 vol%), smoke point (from 12.2 to >50 mm), freezing point (from $-49$ to $<-80$ °C) and net specific energy (41.2–43.7 MJ·kg$^{-1}$), and these properties were compared to standard hydrotreated straight-run Jet A-1 kerosene. The properties of the components studied differed significantly with respect to each other and to the requirements of Jet A-1. Nevertheless, the properties could be well correlated. This provides an opportunity to study possible synergies in blending these components. It was also found that the current methods and instruments used do not always allow a precise determination of the smoke point (>50 mm) and freezing point (<80 °C).

**Keywords:** jet fuel; density; simulated distillation; carbon number; aromatics; n-alkanes; smoke point; freezing point; net specific energy

## 1. Introduction

Jet fuel is a global strategic commodity with a high potential for growth. Consumption is expected to more than triple by 2050 [1]. Currently, the air transport business is responsible for 2.1% of total carbon dioxide emissions or 12% of carbon dioxide emissions from transport [2]. An overview of the standards applicable to jet fuel is available [3,4]. Jet A-1 [5], produced from crude oil straight-run kerosene refined by oxidation of thiols [6] or hydrotreatment, is the most widely used jet fuel in civil aviation. The properties of typical jet fuels have been reviewed in detail [7–9]. The International Air Transport Association (IATA) estimates that SAF's share of jet fuel consumption will be only 0.1% in 2022 [10]. Therefore, it is entirely legitimate that jet fuel should be part of a general effort to reduce harmful emissions from transport originating from petroleum hydrocarbons. Sustainable aviation fuels (SAFs) are now the subject of extensive research and testing and are an important part of strategic plans for the further development of air transport. It is clear that the future role of SAFs will be much more important in this respect than in car transport, where competing alternatives such as electromobility are now dominating.

Several reviews and strategies have been published on all aspects of SAF implementation [11–29], which together include hundreds of references on this topic. SAFs are part of Task 39 of the International Energy Agency (IEA), and the IEA is very active in this regard [30–32]. The use of various biofeeds and biotechnologies dominates these reviews. Based on these reviews, there are other primary issues associated with the implementation of SAFs: availability, costs [33–37] and different quality [38,39] compared to petroleum jet fuel. It is important to note that the currently applicable jet fuel quality standards were derived from the properties

of the petroleum fractions. However, the typical properties of individual SAFs may differ significantly from those of petroleum kerosene. Therefore, SAFs must be qualified for air operations [38]. The drop-in concept plays an important role in overcoming some of the quality problems. The drop-in concept means that SAF can be blended with petroleum-based jet fuel, and the final product does not require changes in infrastructure or equipment. The approved SAF and the maximum concentration in jet fuel are standardized [39]. Jet fuel blends with different compositions have been studied in terms of jet fuel quality [40–42]. Clearly, the most successful SAFs at present are hydrotreated esters and fatty acids (HEFA), which use mainly second-generation feedstocks, especially cooking oils and waste fats [20]. Related technologies were primarily commercialized for the production of hydrotreated vegetable oil (HVO) as a component to diesel fuel [43]— for example, NEXBTL, since 2007 [44,45] in Porvoo, Rotterdam; Singapore oil refineries, and Ecofining, since 2009 [46] in Venice and Gela biorefineries, Livorno; and Sines oil refineries. Each HVO unit can produce from 15 vol% (without additional CAPEX) to 50 vol% (additional CAPEX is required) of HEFA. Moreover, the quality of HEFA is similar to other SAFs based on synthetic i-alkanes produced primarily from the synthesis gas of different origins using Fischer–Tropsch technology followed by the hydrocracking of synthetic crude oil and isomerization of the kerosene fraction [47,48]. Progress in the commercialization of biojet fuel has been discussed [49] and reported [50–54]. From a circular economy point of view, SAFs produced from bulk wastes of petrochemical origin (waste polyolefins [55–60]) and scrapped tires [61] using mature pyrolysis and hydrogenation technologies are very interesting. Regarding the strategy of rapidly increasing jet fuel consumption in a situation of declining crude oil consumption, it is important to address alternative petroleum fractions applicable to this strategy as a basis for the successful implementation of SAFs [62]. These are mainly kerosene from the hydrocracking of vacuum distillates [4,63,64], hydrotreated heavy naphtha from FCC [65], or kerosene obtained instead of the gasoline component by oligomerization and hydrogenation of $C_4$ hydrocarbons [66,67]. The use of these alternative fractions for jet fuel production can be very interesting; however, it will require specific research on additional CAPEX in oil refineries.

The purpose of the research was to analyze the properties of kerosene samples of different origins, produced by different technologies, and with significantly different properties, all of which provide significant potential for increasing jet fuel production. The quality of standard Jet A-1 produced from hydrotreated straight-run kerosene was compared with alternative kerosene fractions produced by hydrocracking and FCC technologies and with SAFs represented by HEFA and hydrotreated kerosene from the pyrolysis of waste polyolefins and scrapped tires. The focus was mainly on the fractional and group composition of the samples and on jet fuel-specific properties such as aromatic and diaromatics content, smoke point, freezing point, and net specific energy. The samples were selected to be within the possible limits of the physical and chemical properties of the alternative fractions currently being considered, researched, and tested for the production of jet fuel. This is important for assessing synergies when blending fractions. It was also examined whether the methods and analytical instruments currently in use are capable of determining the critical properties of the alternative fractions considered for Jet A-1 and whether these properties of samples of very different origin and composition can be correlated as a function of, for example, mean boiling point, n-alkanes and aromatics content.

## 2. Materials and Methods

To monitor the properties of the investigated kerosene samples, the following analytical methods and devices were used: Density—EN ISO 12185 and Anton Paar DMA 4000 density analyzer with an oscillating U-tube (Anton Paar GmbH, Graz, Austria). The measurement was carried out at 15 °C. Distillation—ASTM D86 (A) and NDI 440 Monitoring & Control Laboratories automatic distillation unit (Monitoring & Control Laboratories, Frankenwald, South Africa). Exactly 100 mL of sample was consumed for each analysis. Simulated distillation (SimDist)—ASTM D2887 and the Trace GC Ultra gas chromatograph (Thermo Fisher Scientific Inc., Waltham, MA, USA). Primary chromatographic data were

also used to calculate the approximate content of n-alkanes and hydrocarbon groups with the same number of carbon atoms. Flash point—ASTM D56 (A) and Tanaka ATG-7 automated instrument (Tanaka Scientific Limited, Tokyo, Japan). Exactly 50 mL of sample was used for each analysis. Aromatics—ASTM D6379 (A) and Shimadzu HPLC-RID automated instrument (Shimadzu Corporation, Tokyo, Japan). Aromatics, diaromatics, and polyaromatics were separated on a liquid chromatograph column. Detection was carried out using a refractive-index detector. Approximately 1 g of sample was consumed for each analysis. Smoke point—ASTM D1322 and ATS Scientific Inc. SP 10 automatic instrument (AD Systems, Saint André sur Orne, France). The result was calculated as an average of three measurements. The instrument measures the smoke point from 0 mm to 50 mm. Exactly 20 mL of the sample was consumed for each analysis. Freezing point—ASTM D5972 and Phase Technology FPA-70X automatic instrument (Phase Technology, Richmond, Canada), which measures the freezing point from 20 °C to −80 °C. Approximately 20 mL of sample was consumed for each analysis. Net specific energy measured—ASTM D4809 and LECO AC-350 automated instrument (LECO Corporation, Sant Joseph, USA). The instrument was calibrated with benzoic acid. Using elemental analysis according to ASTM D5291, the hydrogen and water contents were determined, and these values were entered into the instrument. The results of the elemental analysis were used to calculate the $(H/C)_{at}$ ratio. Approximately 0.5 g of sample was consumed for each analysis. Net specific energy calculated—ASTM D3338 using known aromatic content, mean boiling point from ASTM D86 distillation and density. All test methods were applied in accordance with standard test procedures and uncertainty of measurement was lower than repeatability of the corresponding method.

The redistillation of hydrotreated pyrolysis oil samples was performed on a distillation apparatus from Fischer Technology. The 150–250 °C fraction was taken as kerosene.

The following samples were studied: Jet A-1—the hydrotreated straight-run kerosene distilled from light crude oil. It was a commercial refinery product for which a quality certificate was available. A similar sample can be obtained in a number of oil refineries. Therefore, it was used to validate the measured data. This sample represented a substantial part of the jet fuel currently produced [9]. Jet HC—kerosene produced by deep hydrocracking of vacuum distillates from medium-heavy crude oil, of the quality used in the refinery for Jet A-1 production. A similar sample can only be obtained in some oil refineries since it requires adding stabilization, kerosene fraction additivation and separate storage and distribution of jet fuel to the standard hydrocracker unit. This utilization of kerosene provides an interesting opportunity for hydrocracking-type refineries to increase refinery margins [4]. FCC HN—hydrotreated heavy naphtha obtained by fluid catalytic cracking (FCC) of atmospheric residue from light crude oil, subsequent redistillation of FCC gasoline on a 3-cut splitter and hydrotreating of heavy fraction. FCC hydrotreated heavy naphtha is the standard output of the catalytic cracking process. The fraction is commonly used for the blending of mogas and is currently also used for the production of jet fuel as a drop-in component in units of percent. This represents an opportunity to diversify the products from the FCC. PyrTIR—kerosene from scrapped tires pyrolysis oil from an external pilot unit, hydrotreated in a laboratory scale fixed-bed catalytic unit [68,69] with a capacity of 33 g·h$^{-1}$ under severe conditions (360 °C, 10 MPa, commercial hydrotreating NiMo/$\gamma$-Al$_2$O$_3$ catalyst) and redistilled in the laboratory as a fraction with the distillation range of 150–240 °C. PyrPO—kerosene from waste polyolefins pyrolysis oil, treated as described above for the previous sample. Both samples based on pyrolysis oil were prepared specifically for this research. Considering the capacity of the laboratory unit and the yield of kerosene from the hydrotreated product, it was difficult to obtain a sample in the required quantities. Therefore, the focus, in this case, was on the critical and specific properties of jet fuel. Jet fuel from the pyrolysis of waste polyolefins and scrapped tires represents an important opportunity to implement the chemical waste recycling concept and is a major research and development challenge currently. All of the above samples can be considered to be crude oil based. HEFA Cam—biokerosene made from camelina, for which

quality data were available from an independent laboratory because this component was previously used to research drop-in components for jet fuel [40]. HEFA 215—biokerosene provided by a partner laboratory from an undisclosed source as an alternative to HEFA from camelina. Both HEFA samples met the quality requirements of ASTM D7566 [39], which was considered essential for their use in research. HEFA, as an alternative jet fuel, is currently the focus of attention of SAF producers, aircraft manufacturers and major airlines.

The authors of this paper do not have permission to mention the manufacturer of the samples used.

The results presented in this paper were part of a master thesis defended at the University of Chemistry and Technology, Prague, in 2022 [70].

## 3. Results and Discussion

In line with the research objectives described in the introduction, properties important for the evaluation of the studied samples and limiting their yield as jet fuel were summarized and compared with the current Jet A-1 requirements [5] (Table 1). The Table additionally includes the $(H/C)_{at}$ ratio from the elemental analysis (ASTM D5291) and the calculated net energy value (ASTM D3338). Values that did not meet the Jet A-1 requirements are underlined. The data are further discussed in the following figures.

**Table 1.** Key properties of the samples studied.

| Component | JIG Jet A-1 Requirements | Jet A-1 | Jet HC | FCC HN | HEFA Cam | HEFA 215 | PyrTIR | PyrPO |
|---|---|---|---|---|---|---|---|---|
| Density at 15 °C (kg·m$^{-3}$) | 775–840 | 802.3 | 817.3 | 858.8 | 759.5 | 760.7 | 850.9 | 794.8 |
| Distillation (°C) | | | | | | | | |
|   10% distilled (°C) | max 205 | 178.5 | 181.0 | 178.2 | 164.4 | 180.1 | 175.8 | 180.5 [1] |
|   End of distillation (°C) | max 300 | 234.9 | 228.6 | 232.3 | 279.0 | 271.0 | 238.9 | 240.5 [1] |
|   Distillation residue (vol%) | max 1.5 | 1.1 | 1.1 | 1 | 1.1 | 1.4 | 1.2 | [1] |
|   Distillation loss (vol%) | max 1.5 | 0.3 | 1 | 0.3 | 0.1 | 0.2 | 0.8 | [1] |
| $(H/C)_{at}$ | - | 1.928 | 1.882 | 1.550 | 2.177 | 2.172 | 1.708 | 1.972 |
| Aromatics content (vol%) | max 26.5 | 19.7 | 22.7 | 59.7 | 0.3 | 0.0 | 44.2 | 15.9 |
|   Monoaromatics | | 18.5 | 22.6 | 54.2 | 0.3 | 0.0 | 43.8 | 15.7 |
|   Diaromatics | | 1.2 | 0.1 | 5.5 | 0.0 | 0.0 | 0.4 | 0.2 |
| Smoke point (mm) | min 18 | 22.2 | 18.9 | - | >50 [2] | >50 [2] | 12.2 | 26.6 |
|   For naphtalenes > 3 vol% | min. 25 | - | - | 9.3 [3] | - | - | - | - |
| Freezing point (°C) | max −47 | −55.4 | <−80 | <−80 | −57.2 | −49.1 | −80 | −50.1 |
| Flash point (°C) | min 38 | 50 | 53 | 54.5 | 43.5 | 43 | [1] | [1] |
| Net specific energy (MJ·kg$^{-1}$) | min 42.8 | | | | | | | |
|   measured (ASTM D4809) | | 42.8 | 42.7 | 41.2 | 43.3 | 43.7 | 42.2 | 43.0 |
|   calculated (ASTM D3338) | | 43.2 | 43.0 | 42.0 | 44.1 | 44.1 | 42.3 | 43.4 |

[1] Not measured due to lack of sample. ASTM D86 points calculated from SimDist; [2] sample did not smoke; [3] as diaromatics. Values that did not meet the Jet A-1 requirements are underlined.

Distillation is the simplest technology in oil refineries to control product quality. In the case of jet fuel, it is used to control the distillation range, flash point, freezing point in straight-run oil fractions and aromatics in cracking fractions by setting the correct initial and end point of distillation. ASTM D86 is the basis for determining the distillation characteristics of products in oil refineries and was also used for the samples studied (Figure 1).

Since not enough PyrPO sample was available to determine the ASTM D86 distillation, it was substituted in Figure 1 by values calculated from SimDist. The distillation curves of the crude oil-based samples, including PyrTIR and PyrPO, were very similar, the distillation range was relatively narrow (95–5 vol% < 59 °C) and the samples were characterized by a large reserve at 10 vol% distillation temperature and the end of distillation compared to the requirements of JET A-1. The distillation curves of the two HEFA samples differed from the crude oil-based samples and were characterized by a significantly higher range of boiling points (95–5 vol% > 96 °C), therefore, they were steeper.

SimDist (ASTM D2887) provided a more precise distillation characteristic (Figure 2).

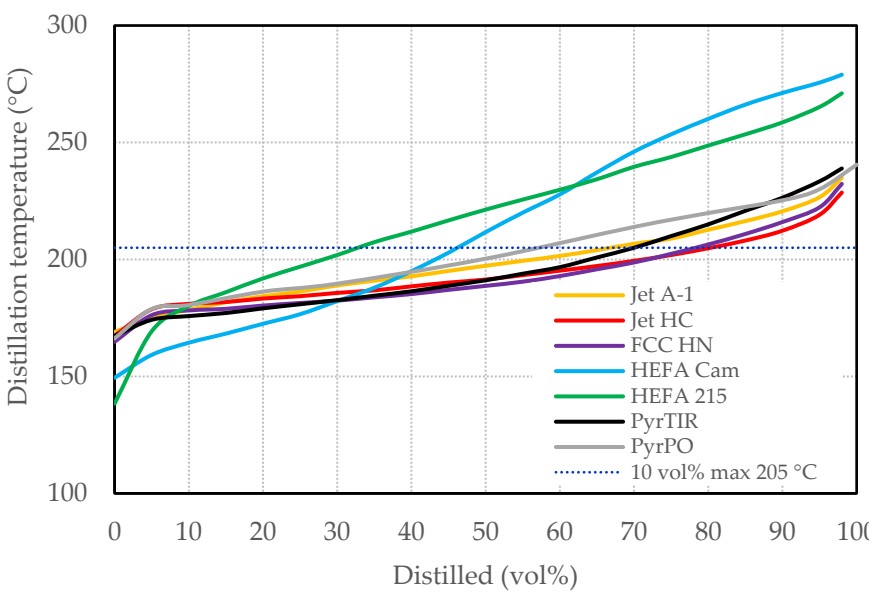

**Figure 1.** ASTM D86 distillation of the samples studied.

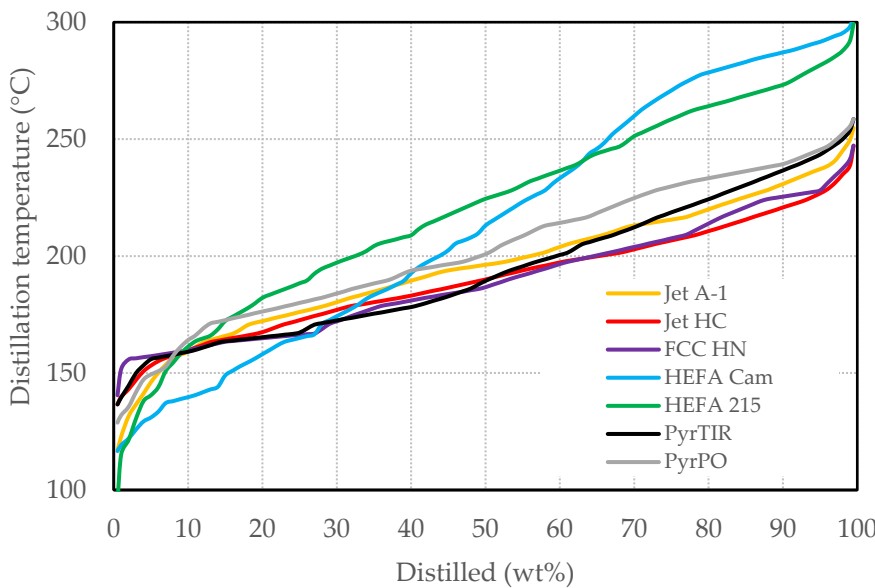

**Figure 2.** SimDist of the samples studied.

As it results from the nature of the SimDist ASTM D2887 method, which allows a better separation of hydrocarbons by boiling point than the ASTM D86 method, the distillation curves were significantly steeper than those from ASTM D86. However, their relationship was similar to Figure 1, i.e., similar for the oil-based samples and clearly different for the two HEFA samples.

The SimDist results allow us to calculate the ASTM D86 distillation curve. Since the samples studied were of very different origin and composition, it was interesting to compare the experimental ASTM D86 results with these calculated values (Table 2).

Table 2 shows that the SimDist-based calculation can be well used to estimate the ASTM D86 distillation of the kerosene samples of very different quality. The biggest differences were found in IBP. The average absolute deviation of both methods did not exceed 5.1 °C.

The SimDist results allow us to estimate the hydrocarbon distribution (Figure 3) and n-alkanes distribution (Figure 4) by carbon number.

**Table 2.** Differences between experimental and calculated ASTM D86 results of the samples studied (°C).

| Sample | Jet A-1 | Jet HC | FCC HN | HEFA Cam | HEFA 215 | PyrTIR |
|---|---|---|---|---|---|---|
| IBP | 8.8 | −4.3 | −11.0 | −0.9 | −10.9 | −6.3 |
| 5 vol% | 1.4 | 2.2 | −1.2 | 3.1 | −3.7 | −2.3 |
| 10 vol% | 2.1 | 5.3 | 2.9 | 3.8 | −0.6 | 0.8 |
| 30 vol% | 3.3 | 4.8 | 5.0 | −1.6 | −1.3 | 4.1 |
| 50 vol% | 2.5 | 3.1 | 2.9 | −2.8 | −1.2 | 1.8 |
| 70 vol% | 3.5 | 5.1 | 3.3 | −1.9 | −1.1 | 1.7 |
| 90 vol% | 4.4 | 5.4 | 6.4 | 2.4 | 1.5 | 5.0 |
| 95 vol% | 4.3 | 6.6 | 8.9 | 2.8 | 1.4 | 5.3 |
| FBP | 0.8 | 3.4 | 7.4 | −1.2 | −3.0 | −0.4 |
| Average absolute deviation (°C) | 3.3 | 4.2 | 5.1 | 2.3 | 2.7 | 3.0 |

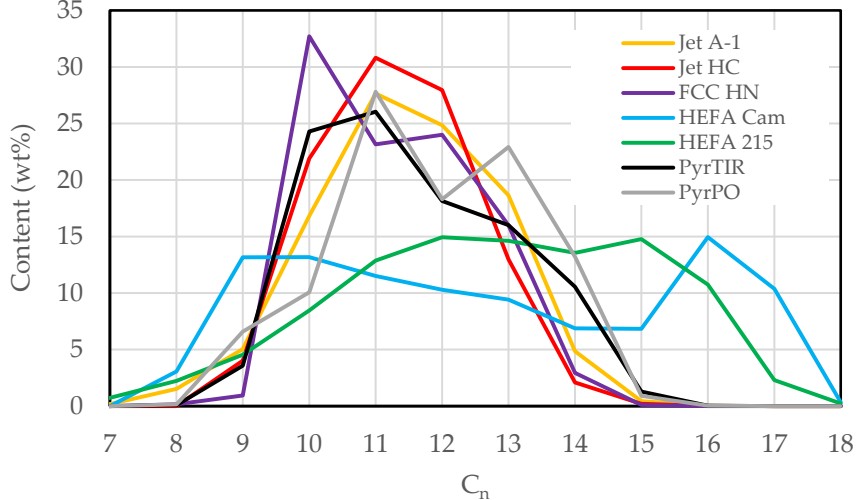

**Figure 3.** Distribution of hydrocarbons in the samples studied by carbon number.

Each of the samples studied has its characteristic imprint in Figure 3, which differed more significantly than the distillation curves. The high concentration of hydrocarbons $C_{10}$–$C_{13}$ was typical for crude oil-based samples. The Jet A-1 sample based on the hydrotreated straight-run kerosene had a very similar hydrocarbon distribution to kerosene from hydrocracking (Jet HC). In contrast, in both HEFAs, a wider range of $C_9$–$C_{16}$ hydrocarbons was present, without a significant maximum. Logically, the narrower the distillation range of the samples studied (Figure 2), the smaller the difference in the carbon numbers of the samples.

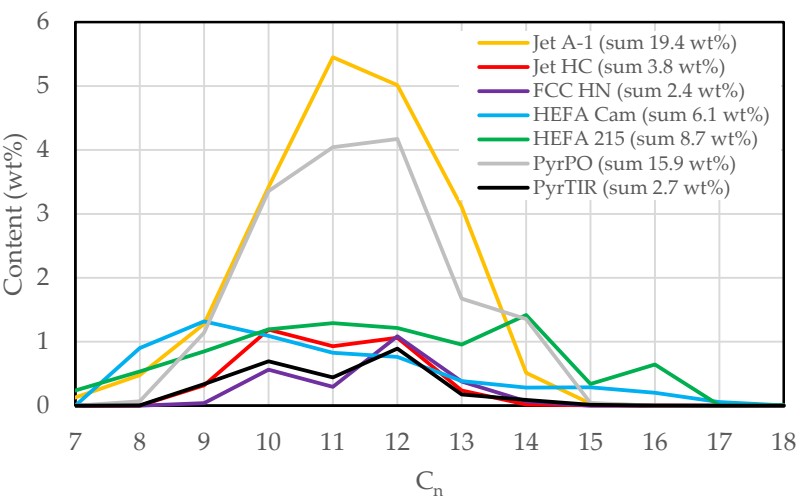

**Figure 4.** Distribution of n-alkanes in the samples studied by carbon number.

The total n-alkane content of each sample is given in the figure legend. It varied substantially from sample to sample. For samples representing the product of cracking technologies, i.e., hydrocracking (Jet HC, HEFA Cam and HEFA 215), catalytic cracking (FCC HN) and pyrolysis (PyrTIR), the n-alkane content was logically low because n-alkanes crack or isomerize. In the waste polyolefins pyrolysis (PyrPO) sample, n-alkanes were a dominant reaction product mainly for polyethylene cracking. For the Jet A-1 sample, the distribution of n-alkanes corresponded to their presence in straight-run kerosene because n-alkanes are not converted during hydrotreatment, and the concentration was the highest of all samples.

The n-alkane content in kerosene is important due to its direct relationship with the freezing point, which determines the temperature of the formation of the paraffin crystals (Figure 5).

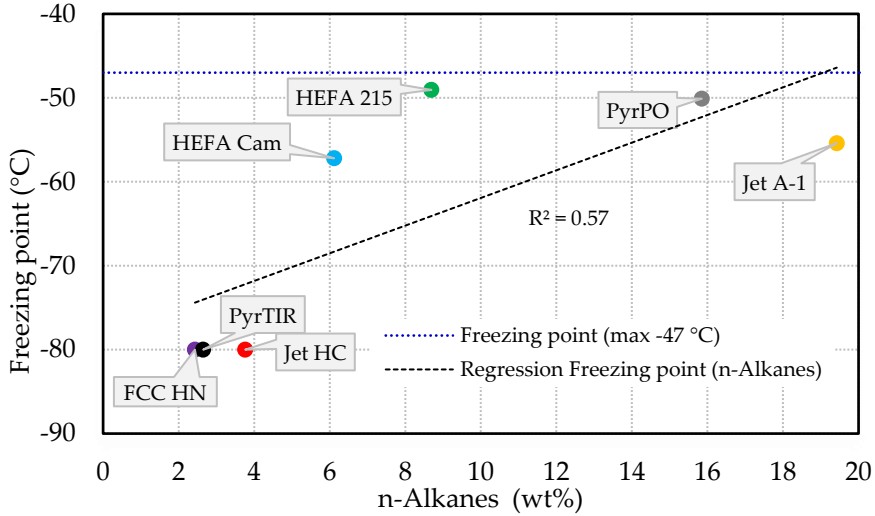

**Figure 5.** Freezing point relation to the n-alkanes content for the samples studied.

All components met the Jet A-1 freezing point requirements (max $-47$ °C, see the dotted line in Figure 5). For the three samples studied (Jet HC, FCC HN, and PyrTIR), the freezing point was not measurable by the method used (ASTM D5972); therefore, the value of $-80$ °C was considered. This may provide an incentive for further development of this method with respect to samples with low concentrations of n-alkanes (<4 wt%) or an extremely low freezing point (<80 °C). On the contrary, both HEFA samples had a relatively high freezing point despite their moderate n-alkane content (<9 wt%). This may be due to the presence of $C_{15}$ and $C_{16}$ n-alkanes in these samples. Although the statement that the freezing point increases with the n-alkane content was generally true for the samples studied, this relationship was not trivial, as shown by the low correlation coefficient $R^2 = 0.57$ for the linear regression in Figure 5. Thus, the n-alkane content was not the only important factor in determining the freezing point of kerosene samples. Aromatics appeared to be another important factor in the behavior of the samples at low temperatures (Figure 6)

The freezing point of the samples decreased with increasing aromatics because the aromatics probably hindered the formation of n-alkane crystals. If aromatics were included as an additional variable in the multivariable linear regression, the value of the correlation coefficient improved significantly ($R^2 = 0.85$). As the alternative components of jet fuel generally represent fractions of very different aromatic contents, this may be a subject of further research.

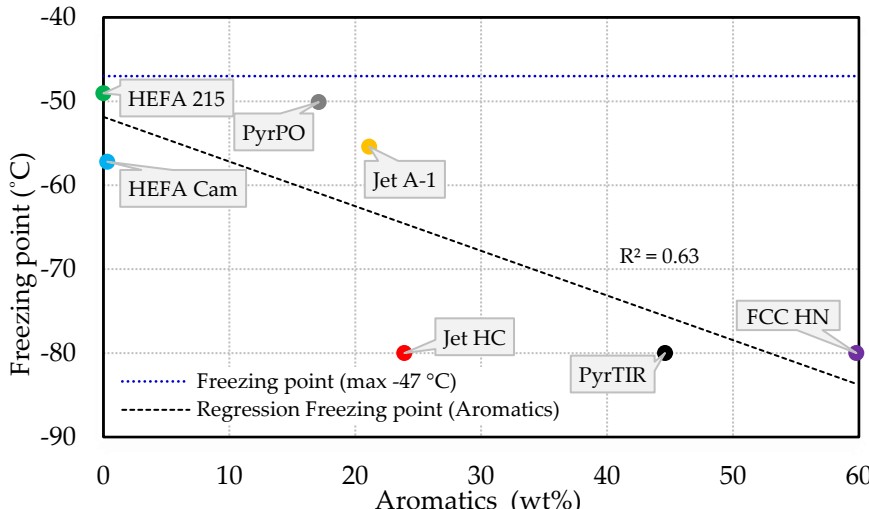

**Figure 6.** Freezing point relation to the aromatics for the samples studied.

Density is an important characteristic of all petroleum fractions. For jet fuel, density requirements are very liberal, which means that the allowed range is very wide (Table 1). For samples of similar origin, it increases with boiling point. However, for samples of different origin and composition, this relationship is more complex and may even be reversed (Figure 7).

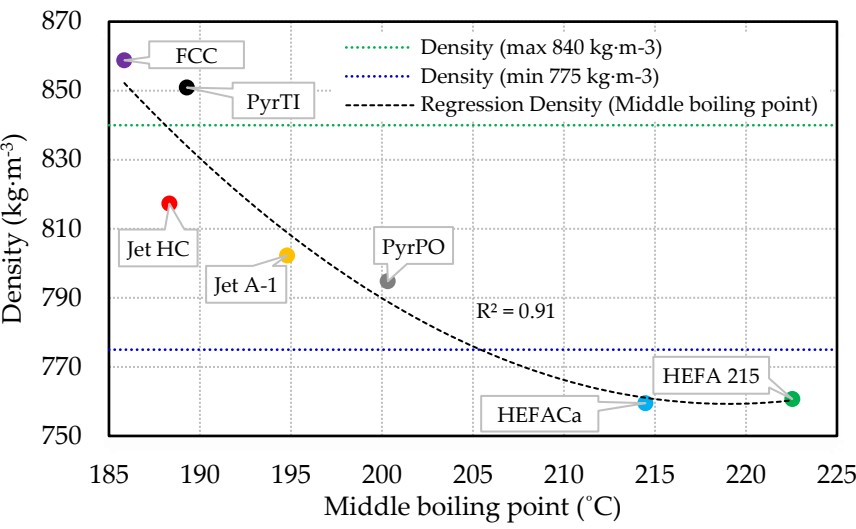

**Figure 7.** Density relation to the middle boiling point of the samples studied.

Although all of the samples studied met the distillation characteristics and the freezing point required for jet fuel, only three samples met, in principle, the basic density requirement. The density of both HEFA samples was too low due to the almost zero aromatic content, while for the PyrTIR and FCC HC samples, the density was too high, based on their high aromatics (see Table 1). Therefore, for the density of the samples studied, the group composition and aromatics were more important than the distillation characteristic (Figure 8).

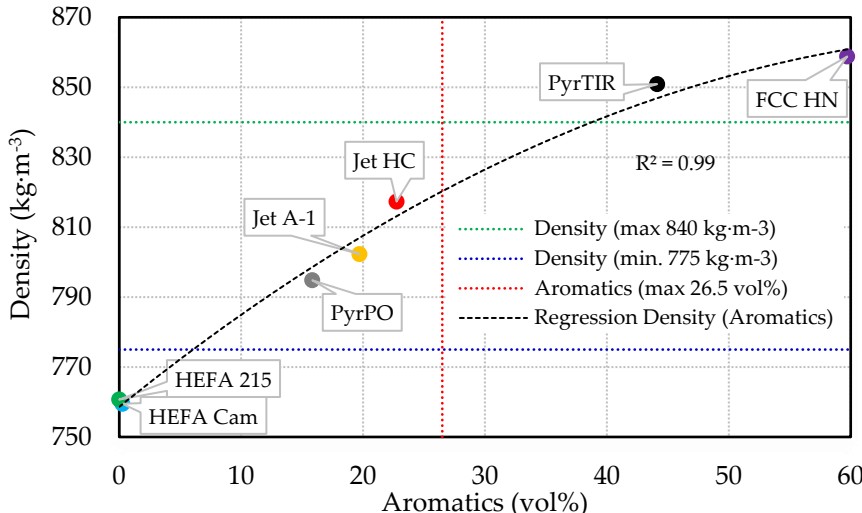

**Figure 8.** Density relation to the aromatics of the samples studied.

For the samples studied, the density relation to the aromatics was almost linear. The density increased with increasing aromatics. From the samples that did not meet the required density, the PyrTIR and FCC NH samples also did not meet the aromatics requirement. This signals that the use of some samples studied in jet fuel may be problematic because they did not simultaneously meet several critical parameters of jet fuel. However, there might be a possibility of blending samples that are on opposite sides of the density and aromatic spectra, which will be the subject of further research.

The aromatics in jet fuel are closely related to another specific qualitative parameter of this product—the smoke point. In general, the smoke point decreases with increasing aromatics [4]. Which of these characteristics limit the production of jet fuel at the oil refinery first always depends on the specific case. This dependence was investigated for the samples studied (Figure 9).

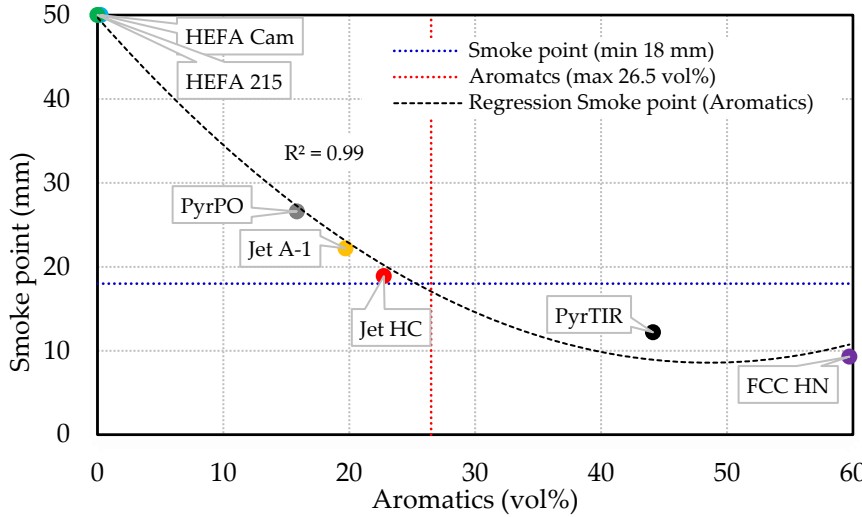

**Figure 9.** Smoke point relation to the aromatics content for the samples studied.

The dependence of the smoke point on the aromatics was clearly different for samples with a concentration of up to 26.5 vol% and with higher aromatics. For samples with aromatics above the JIG requirement for Jet A-1 max 26.5 vol%, the effect of aromatics on the smoke point was less significant. As far as the two HEFA samples were concerned, the aromatics were so low that the smoke point could not be measured. Therefore, the maximum value of the laboratory instrument used (50 mm) was considered in Figure 9.

Only two samples (PyrTIR and FCC HN) did not meet the smoke point of 18 mm. Furthermore, since the FCC HN sample contained 5.5 vol% diaromatics, it should meet the stricter limit of the smoke point minimum (25 mm), and therefore, this sample was significantly outside the JIG requirement for Jet A-1. Based on Figure 9, it can be concluded that there is a nonlinear relationship between the smoke point and aromatics for components with very different aromatics.

In jet fuel, aromatics represent a chemical structure with the lowest $(H/C)_{at}$ ratio. Since hydrogen generally has the highest net specific energy (NSE) per unit mass of all elements, samples with the highest aromatics should have the lowest value of all samples (Figure 10).

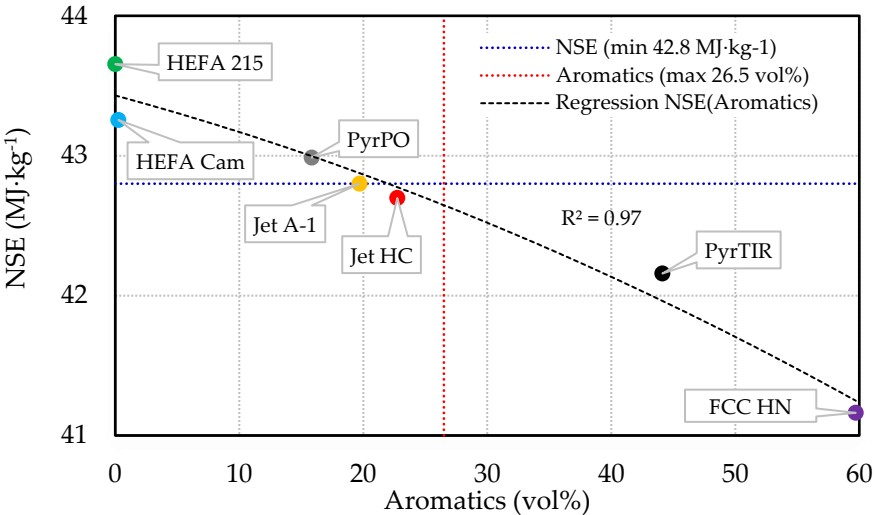

**Figure 10.** Relationship between measured net specific energy ($MJ \cdot kg^{-1}$) and aromatics of the samples studied.

The results confirmed the expectations. The two HEFA samples with zero aromatics and the highest $(H/C)_{at}$ ratio (Table 1) had the highest net specific energy, while the two very aromatic samples (PyrTIR and FCC HN) had the lowest value. The Jet A-1 sample, made from hydrotreated straight-run kerosene, had a net specific energy value at the limit of the requirement ($42.8 \ MJ \cdot kg^{-1}$).

ASTM D3338 provides a correlation to calculate the net specific energy. It was interesting to see how accurate this calculation is for samples with compositions significantly different from standard jet fuel. A comparison of experimental and calculated values yielded a maximum deviation of 2.1%, indicating that this correlation can be used on a wide range of jet fuel compositions (Table 1).

From the point of view of an airline carrier, the net specific energy in a unit volume of jet fuel, which depends not only on the chemical composition but also on the density of the jet fuel, should be more important than the net specific energy in a unit mass (Figure 11).

This dependence is the opposite of the dependence depicted in Figure 10. Thus, the density has a greater effect on the net specific energy per unit volume of jet fuel than the $(H/C)_{at}$ ratio. For example, a unit volume of the sample of standard jet fuel (Jet A-1, density $802.3 \ kg \cdot m^{-3}$, $(H/C)_{at}$ 1.928) contained 4.3% more energy than the HEF Cam sample (density $759.5 \ kg \cdot m^{-3}$, $(H/C)_{at}$ 2.177).

The experimental results obtained for the Jet A-1 and HEFA Cam samples agree very well with the certificates from the independent laboratories available for these samples, which confirmed the precision of the results obtained.

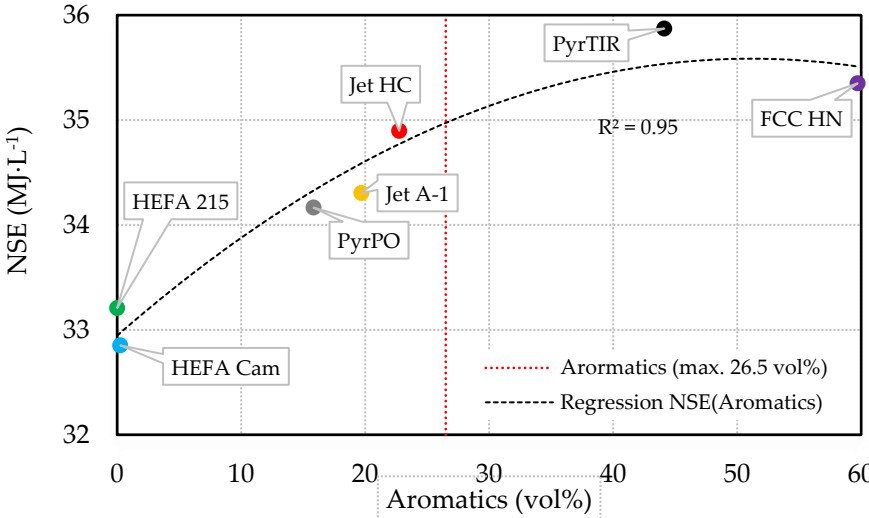

**Figure 11.** Relationship between measured net specific energy (MJ·L$^{-1}$) and aromatics of the samples studied.

## 4. Conclusions

Seven kerosene samples of very different technological origin and composition were studied in terms of Jet A-1 property requirements. The samples represented not only prospective SAFs (HEFA and kerosene from pyrolysis of recycled plastics and scrapped tires) but also interesting crude straight-run kerosene petroleum alternatives, i.e., kerosene produced by deep hydrocracking of vacuum distillates and FCC hydrotreated heavy naphtha. The focus of the study was on the specific and critical properties of these samples related to jet fuel.

Both HEFA samples differed significantly by zero aromatics and non-smoke combustion from the other petroleum-based ones. Four out of the seven samples studied did not meet the current JIG requirements for Jet A-1. These samples were the two HEFA (density < 775 kg·m$^{-3}$), FCC hydrotreated heavy naphtha and hydrotreated kerosene, from the pyrolysis of scrapped tires (for both density > 840 kg·m$^{-3}$, aromatics > 26.5 vol%, smoke point < 18 mm and net specific energy < 42.8 MJ·kg$^{-1}$). Thus, the critical properties of the samples were density, aromatics, smoke point and net specific energy. In contrast to the current prevailing practice of producing jet fuel by simple additivation of a single fraction, it will be necessary to blend the studied alternative components with straight-run kerosene (drop-in concept) or with each other, i.e., similar to the way other motor fuels are now produced. Some of the properties of the samples studied differed significantly from those of the petroleum fractions currently used (freezing point < −80 °C, smoke point > 50 mm, density significantly outside the required range of 775–840 kg·m$^3$), so it will be necessary to consider modifications to the existing standards and the modification of analytical methods to determine their properties in a wider range (freezing and smoke point). This will be extremely important to properly evaluate the benefits of alternative components in refinery optimization models. The results of the measurements inspired further research aimed at blending alternative petroleum components with SAFs and components that lie on the opposite sides of the spectrum of the properties studied, such as HEFA and FCC hydrotreated heavy naphtha.

**Author Contributions:** Conceptualization, H.K. and J.H.; methodology, P.Š.; software, P.Š.; validation, H.K., J.H. and P.Š.; formal analysis, H.K. and P.Š.; investigation, P.Š.; resources, H.K. and J.H.; data curation, H.K.; writing—original draft preparation, H.K.; writing—review and editing, J.H. and P.Š.; visualization, H.K.; supervision, P.Š.; project administration, H.K.; funding acquisition, P.Š. All authors have read and agreed to the published version of the manuscript.

**Funding:** This research was funded by the Ministry of Education, Youth and Sports of the Czech Republic from the institutional support of the research organization (CZ60461373).

**Data Availability Statement:** The data presented in this study are available on request from the corresponding author.

**Conflicts of Interest:** The authors declare no conflict of interest.

## Nomenclature

| | |
|---|---|
| ASTM | American Society for Testing and Materials, West Conshohocken, PA, USA |
| CAPEX | Capital Expenditures |
| EN | Euro Norm, Office for Official Publications of the European Communities, Luxembourg, Luxembourg |
| FBP | Final Boiling Point |
| FCC | Fluid Catalytic Cracking |
| HEFA | Hydrogenated Esters and Fatty Acids |
| HN | Heavy Naphtha |
| HVO | Hydrogenated Vegetable Oil |
| IATA | International Air Transport Association |
| IBP | Initial Boiling Point |
| IEA | International Energy Agency |
| ISO | International Organization for Standardization, Geneva, Switzerland |
| JIG | Joint Inspection Group, Cambourne, Great Britain |
| NSE | Net Specific Energy |
| SAF | Sustainable Aviation Fuels |
| UCT | University of Chemistry and Technology Prague |

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
