# Peer review of "Properties of Selected Alternative Petroleum Fractions and Sustainable Aviation Fuels"

_processes, doi:10.3390/pr11030935_

Round 1

Reviewer 1 Report

The manuscript requires modification before it can be considered for publication in the Journal. Here are a few of the guidelines the authors should follow when revising the manuscript:

·         The analysis in the Abstract is predominantly qualitative, and the results of the quantitative analysis are absent. It is suggested that pertinent information be included.

·         The abbreviation table should be included in the manuscript.

·         The introduction section contains very few references and should be reviewed for relevant papers in the field.

·         It would be good to have a short introduction about the results and discussion.

·         Deep and mechanical discussions are necessary to explain the given results.

·         Include quantifiable results and achieved outcomes in the conclusion.

·         Some consideration of future research needs to be included in the conclusion section.

Author Response

See please annexed word file.

Reviewer 2 Report

 This is an interesting work, but the figure needs to be modified. The line color is too close to be seen clearly.

Author Response

See please the annexed word file.

Reviewer 3 Report

Authors tentatively reported an investigation on kerosene like fules as possible replacement of Jet A-1 . The all paper presetn several major flaws and critical methodological issues listed as follow: 

line 30: Please make explicit teh acronym IATA

Section 2: It is totally unreadable. This is core of any scientific paper that allow to other to replicate your experiments. This section must e totally rewrote. It is totally missed any detailed description about teh orgin of the fuels.

Table 1: Data are reported as mere numbers not as scientifc values. A scientific value must be reported together with its uncertanty.

lines 163-164: "Because not enough PyrPO sample was available to determine the ASTM D86 distil- 164 lation, it was substituted in Figure 1 substituted by values calculated from SimDist" this is a problem of the authors. If authors want to include PyrPO they must have a sufficent quantity.

Figure 3-11: Data reported in figure 3-11 are totally meaningless wihtout uncertanties. Authors cannot comapare number but only scientific values byusing the appropriated statistical tools.

The statical analysis of teh data is totally missed. In this way, the manuscript is only a poor written technical report far away form the minimum standard of publication in any reputated journal.

I cannot endorse its publication of the present paper and i strongly discourage the author to submitt a scientific research witou use the appropriated statistical tools for the critical and rational comparison of the data.

Author Response

See please the annexed word file.

Round 2

Reviewer 1 Report

The author incorporated all of the reviewer's suggestions. I may recommend that this article be published in a journal.

Reviewer 3 Report

Authors answer to my questions in a rigorous way. The answer were disgracefully not sufficent to justify their claim. 

". Uncertainty of measurement was lower than repeatability of the corresponding method"

This statement is totally useless. If the uncertanty is as an example 1% you results sheould be presented as 100 +/- 1 

Authors did not do that. Claim that a value is different from another one must be suported by t-test, trends must be supported by ANOVA.

Accoridngly, the scientific value of this work is still below the mark for pubblcaiton even authors answer to the other quesitons in a satisfactionary method.